# Evaluation of Changes to the Oral Microbiome Based on 16S rRNA Sequencing among Children Treated for Cancer

**DOI:** 10.3390/cancers14010007

**Published:** 2021-12-21

**Authors:** Patrycja Proc, Joanna Szczepańska, Beata Zarzycka, Małgorzata Szybka, Maciej Borowiec, Tomasz Płoszaj, Wojciech Fendler, Jędrzej Chrzanowski, Małgorzata Zubowska, Małgorzata Stolarska, Wojciech Młynarski

**Affiliations:** 1Department of Pediatric Dentistry, Medical University of Lodz, Pomorska 251, 92-213 Lodz, Poland; joanna.szczepanska@umed.lodz.pl; 2Department of Microbiology and Medical Laboratory Immunology, Medical University of Lodz, Pomorska 251, 92-213 Lodz, Poland; beata.zarzycka@umed.lodz.pl (B.Z.); malgorzata.szybka@umed.lodz.pl (M.S.); 3Department of Clinical Genetics Medical, University of Lodz, Pomorska 251, 92-213 Lodz, Poland; maciej.borowiec@umed.lodz.pl (M.B.); tomasz.ploszaj@umed.lodz.pl (T.P.); 4Department of Radiation Oncology, Dana-Farber Cancer Institute, Boston, MA 02215, USA; wojciech.fendler@umed.lodz.pl; 5Department of Biostatistics and Translational Medicine, Medical University of Lodz, Mazowiecka 15, 92-215 Lodz, Poland; jedrzej.chrzanowski@umed.lodz.pl; 6Departments of Pediatrics, Diabetology, Endocrinology and Nephrology, Medical University of Lodz, Sporna 36/50, 91-738 Lodz, Poland; malgorzata.zubowska@umed.lodz.pl; 7Department of Pediatrics, Oncology & Hematology, Medical University of Lodz, Sporna 36/50, 91-738 Lodz, Poland; malgorzata.stolarska@umed.lodz.pl

**Keywords:** oral microbiome, cancer patients, children, 16S rRNA sequencing

## Abstract

**Simple Summary:**

Childhood cancer survivors suffer from many oral complications during and after primary therapy. Our study focuses on changes in the oral microbiome of cancer survivors. Using 16S rRNA sequencing, we observed global and distinct changes in oral microbiome associated with a patient’s age and therapy duration, but not antibiotic therapy or cancer type. Observed changes in the oral microbiome could differentiate patients at higher risk of long-term oral complications.

**Abstract:**

A child’s mouth is the gateway to many species of bacteria. Changes in the oral microbiome may affect the health of the entire body. The aim of the study was to evaluate the changes in the oral microbiome of childhood cancer survivors. Saliva samples before and after anti-cancer treatment were collected from 20 patients aged 6–18 years, diagnosed de novo with cancer in 2018–2019 (7 girls and 13 boys, 7.5–19 years old at the second time point). Bacterial DNA was extracted, and the microbial community profiles were assessed by 16S rRNA sequencing. The relative abundances of *Cellulosilyticum* and *Tannerella* genera were found to significantly change throughout therapy (*p* = 0.043 and *p* = 0.036, respectively). However, no differences in the alpha-diversity were observed (*p* = 0.817). The unsupervised classification revealed two clusters of patients: the first with significant changes in *Campylobacter* and *Fusobacterium* abundance, and the other with change in *Neisseria*. These two groups of patients differed in median age (10.25 vs. 16.16 years; *p* = 0.004) and the length of anti-cancer therapy (19 vs. 4 months; *p* = 0.003), but not cancer type or antibiotic treatment.

## 1. Introduction

The emergence of new molecular biology techniques, such as high-throughput sequencing techniques, has completely changed the face of clinical microbiology. Their use in metagenomics research has also increased the speed and accuracy of diagnosis [1] and revealed changes in the bacterial composition throughout the disease [2,3]. One of the latest strategies involves the use of targeted sequencing for the bacterial 16S rRNA gene. This method has already gained many positive results in examining microbiota changes in such regions as the lungs, vagina, urinary tract, or gut [4,5,6,7]. In the Human Microbiome Project (HMP), led by four sequencing centers (the Broad Institute, Baylor College of Medicine, Washington University School of Medicine, and the J. Craig Venter Institute) and the Data Analysis and Coordination Center (DACC), microbial communities of 300 healthy individuals were characterized by 16S rRNA sequencing [8]. The overall goal was to determine if there was a healthy core microbiome, so samples were collected at multiple time points from several different places on the human body: nasal passages, mouth, skin, gastrointestinal tract, and genitourinary tract.

The results of 16S rRNA sequencing indicated a close connection between the microbiota in different compartments of an organism, which were formerly treated as independent niches with their microbiomes. Several recent studies have focused on the immunological interaction between the gut and lungs and found that the composition of their microbiomes is influenced by the diseases or homeostasis of the entire organism [5]. It was also found that bacteria could penetrate from one anatomical region to another, changing the composition of the microbiome in even anatomically distinct regions. For example, pathogenic bacteria typical for the oral cavity (*Fusobacterium nucleatum* and *Porphyromonas gingivalis*) can be found in the colon of patients suffering from periodontitis. These bacteria are highly pathogenic and can seriously alter the composition of the microbiome in the whole gastrointestinal tract, which may lead to chronic inflammation and even to colorectal cancer tumorigenesis [7]. A relationship between oral microbiota and cancer was also observed in another large study, in which the presence of oral bacteria *Porphyromonas gingivalis* and *Aggregatibacter actinomycetemcomitans* were associated with a higher risk of pancreatic cancer [9].

These relationships may also have an influence in situ as the composition of the oral microbiome is responsible for the health of the oral cavity itself, as the oral cavity is the first part of the gastrointestinal tract, and in adults has a microbiome of over 700 microorganisms, including bacteria, fungi, and viruses [10]. Bacteria residing in the oral cavity of a newborn come from people in their immediate surroundings. Colonization takes place in stages of child development to achieve some stabilization in adulthood. This balance may be disturbed by childhood diseases, such as early childhood caries, celiac disease, autism, Henoch-Schönlein purpura, appendicitis in children, inflammatory bowel disease, and obstructive sleep apnea syndrome in children [10].

Changes in the oral microbiome may also be of particular importance in patients with childhood malignant neoplasms who have a number of oral complications like caries, gingivitis, stomatitis, dry mouth, and candidiasis [11]. Patients receiving chemotherapy also had a different microbiome composition than healthy controls, which is a potential risk factor for the development of oral mucositis [11,12]. It was concluded that childhood cancer therapy might affect tooth development, saliva function, craniofacial development, and temporomandibular joint function, exposing some children who have survived cancer to an increased risk of oral and dental problems [13,14].

However, the role of the oral microbiome in cancer treatment and healing remains unclear, and relatively few studies have evaluated the oral microbiome of cancer patients with the use of 16S rRNA sequencing. The present study examines whether the salivary microbiome of childhood patients is altered over the course of anti-cancer chemotherapy.

## 2. Materials and Methods

### 2.1. The Cohort

A total of 71 children newly diagnosed with various types of cancer at the Department of Pediatrics, Oncology & Hematology, between March 2018 and February 2019, were approached to participate in the study. The schema used for participant selection is presented in Figure 1. The inclusion criteria comprised the following: age below eighteen years, newly diagnosed cancer before any treatment, no primary or secondary immunodeficiency identified, and no antibiotics treatment within six months prior to the study.

A total of 30 patients did not meet the inclusion criteria or refused to take part. Therefore, the first saliva sample was taken from 41 children (26 boys and 15 girls) aged 6–18 years. During the study, six patients died, eight had a relapse, four withdrew, and three missed the second sampling. Therefore, a second sample was taken from 20 participants: 7 (35%) girls and 13 (65%) boys. Paired samples from these 20 children were included in the statistical analysis. The patients suffered from the following cancer types: brain tumor (*n* = 4), leukemia (*n* = 4), Hodgkin’s lymphoma (*n* = 4), non-Hodgkin’s lymphoma (*n* = 3), Rhabdomyosarcoma (*n* = 2), neuroblastoma (*n* = 1), ovarian tumor (*n* = 1), and tumor of the testis (*n* = 1).

The median age was 13.79 years at the first saliva collection and 15.58 years during the second collection. The median duration of cancer treatment was 10.5 months (IQR: 4.5–24 months); (Table 1). Detailed clinical characteristics is provided in Appendix A.

Written consent to take part was collected from parents or guardians, as well as from the child, before participation. The study was approved by the Bioethics Committee of the Medical University of Lodz (IRB number: RNN/37/13/KE).

### 2.2. Saliva Collection and DNA Isolation

The first saliva sample was collected on the first day after cancer diagnosis, before any anti-cancer treatment. The second saliva sample was taken when the anti-cancer treatment was finished, i.e., three weeks after the last anti-cancer drug was given, corresponding to a median of 22 (19–24) days after oncological treatment.

The non-stimulated saliva collection process supervised by a nurse took at least 15 min. A collection of 4 mL of saliva into a 15 mL sterile plastic vial (sputum tube, Sarstedt, Germany) was performed in the morning: 2–4 h after breakfast and teeth brushing. Saliva samples were stored immediately at −20 degrees Celsius for further genomic DNA extraction. Bacterial DNA was extracted from frozen 2 mL clinical saliva samples using 200 U mutanolysin and 50 µL lysozyme with a Genomic Midi AX DNA Extraction Kit (A&A Biotechnology, catalog number: 895-20-M1, Gdynia, Poland).

### 2.3. Library Preparation and Sequencing

Microbial community profiles were assessed by sequencing the 16S rRNA gene. The first step was the amplification of the V3 and V4 variable fragments of the analyzed gene according to the manufacturer’s protocol (Illumina, San Diego, CA, USA). Primers with overhanging adapters compatible with Illumina indexes and sequencing adapters in paired-end sequencing techniques were used. Kapa HiFi polymerase (Roche, Cape Town, South Africa) was used to amplify a fragment with a mean length of 464 bp. Next, the specificity of the obtained products was evaluated in an agarose gel and then purified on AMPure XP magnetic beads (Beckman Coulter, Brea, CA, USA). The indexing reactions were also carried out using Kapa HiFi polymerase (Roche, Cape Town, South Africa) with the Nextera XT dual-index set (Illumina, San Diego, CA, USA). The concentration of the obtained libraries was determined using the Qubit 2.0 device (Thermo Fisher Scientific, Waltham, MA, USA) and pooled in equal concentrations. The resulting library was sequenced on the MiSeq platform (Illumina, San Diego, CA, USA) using a MiSeq Reagent Kit v3 (600 cycles).

### 2.4. NGS Data Processing

Raw sequencing data from the MiSeq device were uploaded to the Galaxy web platform [15] and were analyzed using the public server at usegalaxy.org. FASTQ format files were unified to Sanger FASTQ encoding with the FASTQ Groomer tool [16]. Paired-end reads were first merged using the FLASH tool [17], then the Trimmomatic algorithm was used to remove adapters and low-quality reads (below Q20 value) [18]. Operational taxonomic units (OTUs) were assigned by the Kraken algorithm [19] and filtered by the classification confidence score at a level of 0.05. Read counts for each taxonomy level were extracted into tables, and the percentage abundance of each identified bacterial taxon was calculated. Only those taxonomic units that constituted at least 1% of mappable reads in at least one of the examined patients were considered for further analysis.

### 2.5. Data Analysis

#### 2.5.1. Alpha and Beta Diversity

The compositional diversity of the microbiome in the samples was analyzed at the genus level on the web Microbiome Analyst platform [20]. The Chao1 algorithm and the paired t-test were used to determine the alpha-diversity of the microbiome and the statistical significance of any differences. Beta diversity was visualized using non-metric multidimensional scaling (NMDS), and the statistical significance was tested by the PERMANOVA method.

#### 2.5.2. Visualization of Data

Hierarchical Clustering and Heatmap were created using a Ward clustering algorithm and Euclidean distance. The results were visualized as a dendrogram using the Ward clustering algorithm and Bray–Curtis index distance measure; the dendrogram was created using the web Microbiome Analyst platform [16]. Changes in raw abundance before and after anticancer therapy were presented using stacked-bar plots using ggplot2 package in R. Differences in relative abundances between time-points in groups were obtained with k-means clustering and plotted as chard plots using the cyclize package in R.

#### 2.5.3. Statistical Analysis

Changes in the relative abundances were tested with Wilcoxon signed-rank. Patterns of change in oral microbiota were investigated using a k-means clustering algorithm. Differences between relative abundances between time-points were compared using Wilcoxon signed-rank test. Statistical analysis was performed using R-studio, Statistica 13.1 (TIBCO Software, Palo Alto, CA, USA), and Microsoft Excel software. Two-tailed p-values lower than 0.05 were deemed statistically significant. Due to the small study group, no multiple comparisons correction was applied.

## 3. Results

### 3.1. 16S rRNA Sequencing and Microbiome Abundance Analysis

The number of all obtained reads assigned to bacteria was 5,060,942, which translated into a mean value of 126,524 reads per single analyzed sample. The detailed analysis focused mainly on the genus level because it was the lowest level that had a high percentage of assigned reads, with a mean value of 89%. In the analyzed material, 20 different types of bacteria were identified, with the most abundant being *Streptococcaceae* 46.4%, *Prevotellaceae* 12.6%, and *Veillonellaceae* 9.9%, which together accounted for 68.9% of all identified OTUs.

From investigated bacteria, *Tannerella*, *Candidatus Kinetoplastibacterium*, *Lactobacillus*, *Gardnerella*, and *Cellulosilyticum* were determined to have a detection threshold below 0.001% in more than 50% of samples. However, because the study’s main aim was to determine the change in microbiome before and after anticancer therapy, it was decided not to exclude these genera, as they may be clinically relevant, despite their low prevalence (Figure 2b). Statistically significant differences were observed for the relative abundances of *Cellulosilyticum* (88% increase in relative abundance; *p* = 0.043) and *Tannerella* genera (97% increase in relative abundance; *p* = 0.036). Both genera, however, were very rare in the evaluated material. A trend towards significance was also noted for *Bifidobacterium*, *Prevotella*, and *Kribbella* (Table 2).

Alpha-diversity between time points (before and after anticancer therapy) reflects a change in microbiome diversity due to the treatment. However, the observed shift in alpha-diversity was not significant (*p* = 0.9456 for Chao1, 0.7303 for ACE, 0.8440 for Shannon, 0.5297 for Simpson, Figure 2a), indicating conserved microbiome diversity throughout the anticancer therapy (see also Appendix A).

### 3.2. Differences in Microbiome in Patients before and after Cancer Treatment

Despite a non-significant change in alpha-diversity metrics in the course of anticancer therapy, relative changes in genera abundance were further investigated. The microbiological profile of each patient, before and after anticancer therapy, is demonstrated in Figure 3a. For selected genera (*p* < 0.15, Table 2), overall changes in relative abundances were presented using stacked bar plots (Figure 3b–f).

### 3.3. Patterns of Microbiome Changes during Cancer Treatment Is Associated with Clinical Outcomes

To investigate the patterns of changes within patients, we applied k-means unsupervised clustering. Two clusters were identified, with significantly different changes in microbiome. Group median values and inter-quartile ranges for genera significantly altered by treatment in at least one group were provided in Table 3. For Cluster 1 we observed a 62% increase in *Campylobacter* and an 80% increase in *Fusobacterium* abundance. As for Cluster 2, there was a 242% decrease in Neisseria abundance.

The characteristics of changes due to anticancer therapy in the clusters are illustrated via circus plots (Figure 4a,b). To focus on the specific changes, selected genera are presented as a stacked bar plot (Figure 4c–j).

These two clusters differed with the age of patients at the diagnosis (*p* = 0.0049) and with the length of their therapies (*p* = 0.0038); (Table 4 and Figure 5a,b).

Although the length of antibiotic therapy was not significantly different between clusters, an abundance of specific bacteria correlated with the duration of antibiotic treatment. For the *Gardnerella*, we observed a strong positive correlation between the length of antibiotic therapy and its relative abundance after anticancer therapy (R = 0.4729, *p* = 0.0353, Figure 6a). Moreover, Cluster 2 presented a significant correlation between the duration of antibiotic therapy and change in relative abundance pre-post anticancer therapy for *Gardnerella* (R = −0.7280, *p* = 0.0260, Figure 6b) and *Prevotella* (R = 0.7699, *p* = 0.0152, Figure 6c).

## 4. Discussion

Anticancer therapy among pediatric patients consists of multidrug regimens and generates severe immunosuppression; these significantly increase the risk of infection and antibiotic use and, thus, affects the microbiota profile of the patient. Our results show that treatment of pediatric malignancies impacts the salivary microbiota. The length of intensive chemotherapy increases the risk of infection or empirical antibiotic treatment. Interestingly, unsupervised clustering analysis found these clusters to be linked with certain clinical features of the patients, including age at diagnosis and length of cancer therapy. This suggests that saliva microbiota might be modified by anticancer treatment in childhood.

While some changes in components were significant and substantial heterogeneity was observed between the patients, others generally remained constant despite the treatment and immunological suppression. The major component of the microbiome was constituted by *Streptococcus*, which was found to constitute 46.4% of the salivary microbiome. This seems to be in line with the prevalence of the genus in the Polish population and begins with the relationships among the mode of delivery (natural or C-section), the oral microbiome pattern of the infant, and the similarity of commensal bacteria inhabiting the oral cavity of the mother and her placenta [21,22]. *Streptococcus* was also found to be the predominant genus in a pre-dentate mouth (62.2%), later connected with the initiation of dental caries [23]. Other genera usually found to be abundant in the infant’s mouth are *Veillonella*, *Neisseria*, *Rothia*, *Haemophilus*, *Gemella*, *Granulicatella*, *Leptotrichia*, and *Fusobacterium*. This differs from the oral microbiota of adults, in whom *Haemophilus*, *Neisseria*, *Veillonella*, *Fusobacterium*, *Oribacterium*, *Rothia*, *Treponema*, and *Actinomyces* predominated [23]. In our group of children, aged between 6 and 18 years, the second and third most abundant genera were *Prevotella* and *Veillonella* while the remaining comprised 31.1% of the microbiome.

The adult microbiome shows greater variety than those found in children, with the greatest changes taking place up to the age of three years [22]. Teeth gradually erupt, occlusions form, and the diet becomes more complex. As the process of immune postnatal development is closely related to microbial exposure in the gut, diet plays a crucial role in its proper development [24]. In general, the diet alters the structure and function of the microbiome throughout the gastrointestinal tract, which in some cases may lead to a decrease in species richness and promote the proliferation of pathogenic bacterial taxa [25]. Such complications appear when a child suffers from a condition that impairs proper nutrition, such as celiac disease or inflammatory bowel disease [26,27]. Indeed, during cancer therapy, children may often require changes in nutrition or even a need for parenteral feeding [28]. It has also been observed that oncological treatment may influence the outcome of therapy by causing deviations in gut microbiota composition [29].

In the investigated group, a significant difference between the two time-points was found only for *Cellulosilyticum* and *Tannerella*. *Cellulosilyticum* is a newly described genus that could efficiently hydrolyze cellulose, and it is mostly identified in the rumen of ruminantia [30]. In contrast, *Tannerella* is quite commonly detected in the oral cavity of humans and its presence is widely connected with periodontitis and loss of teeth in adults [31]. It was also found that *Tannerella* was overrepresented in the tongue coating in patients with pancreatic head cancer [32].

The abundance of *Prevotella* bacteria was also found to decrease over treatment. This genus is known to be associated with periodontal inflammation, and its abundance has been found to increase in patients with asthma or chronic obstructive pulmonary disease [33]. A trend toward significance was also noted in the abundance of *Bifidobacterium*, whose fraction increased two-fold after the treatment. These bacteria colonize the gastrointestinal tract in children and they are believed to have a range of health-promoting properties [34].

Finally, *Kribbella* also demonstrated considerable changes in the number during therapy. The genus is usually isolated from the soil or plants [35]. This is the first confirmation of its constant presence in the oral microbiome of humans.

To identify the existence of specific microbiome patterns, the patients were clustered with regard to the observed changes in oral microbiome. Two clusters were identified, characterized by changes in *Campylobacter* and *Fusobacterium* (Cluster 1), or *Neisseria* (Cluster 2).

*Campylobacter* is the most common cause of diarrheal illness, including acute enteritis, extraintestinal infections, and post-infectious complications [36], and its increased presence can be easily understood in the microbiome of children treated from cancer. In contrast, while *Fusobacterium* is considered a part of the normal microbiome of the oral cavity, gastrointestinal tract, and female genital tract, it is believed to potentially cause acute otitis media in small children [37]. Increases in the relative abundance of *Fusobacterium* were also confirmed in the colon of colorectal cancer patients [38] and several studies indicate a relationship between *Fusobacterium*, periodontal disease, and oral cancer in adults [39,40].

Regarding Cluster 2, the patients demonstrated a significant decrease in the abundance of *Neisseria*. Some *Neisseria* species are invasive pathogens inhabiting mucosal surfaces in the upper respiratory and genito-urinary tracts, but most are considered part of the commensal microbiome, which is needed to prevent infection in the oropharyngeal area [41]. However, it has been shown that high levels of *Fusobacterium periodonticum* and low levels of *Neisseria mucosa* are specific risk factors for pancreatic cancer [42].

The patients from Cluster 2 were older than children from Cluster 1 and their cancer therapy length was significantly shorter. No significant differences were observed in patients with different durations of antibiotic therapy or taking steroids. However, the one whose abundance was related to the length of antibiotic treatment was *Gardnerella*. When it comes to cluster analysis, there are relationships between the duration of antibiotic therapy and changes in the pre-post percentage of a given bacterium. A short course of antibiotics was correlated with an increase in *Gardnerella* abundance in Cluster 1, but the longer the course, the smaller the change or even increase in abundance.

This is in line with the results of others, which show that in men treated with antibiotics for urethritis, the *Gardnerella* abundance in oral saliva did not change significantly [43].

This was also the case for *Prevotella*, even though its change in abundance was not significant after clustering, the *Prevotella* count in Cluster 2 patients was related to the duration of antibiotic treatment and was proportional to the duration of antibiotic treatment.

There were also no changes in patients with different types of cancer or the presence of caries. The non-significance may be related to the small number of patients in the groups and this can generally be considered the main limitation of the study. As such, our results should be interpreted tentatively and further studies with a larger sample size are needed. Even so, our study is the first to alter the oral microbiome in childhood cancer survivors using 16S rRNA genotyping.

## 5. Conclusions

Our results clearly indicate the feasibility of measuring changes in oral microbiome in the course of chemotherapy and provide insight into the changes of oral microbiome in children in course of therapy for cancer. However, further studies involving greater cohorts are necessary, and these should include factors such as dietary habits and proven infections throughout the course of chemotherapy.

## Figures and Tables

**Figure 1 cancers-14-00007-f001:**
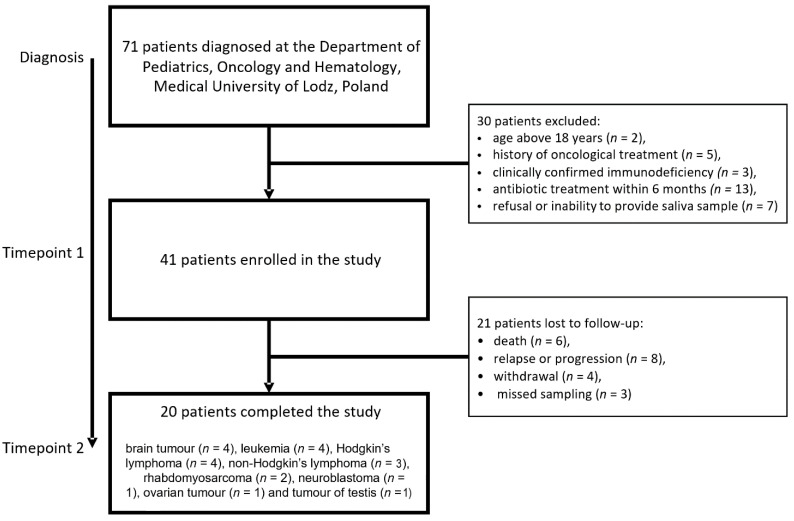
Plan of the study—number of patients enrolled in each stage of the study.

**Figure 2 cancers-14-00007-f002:**
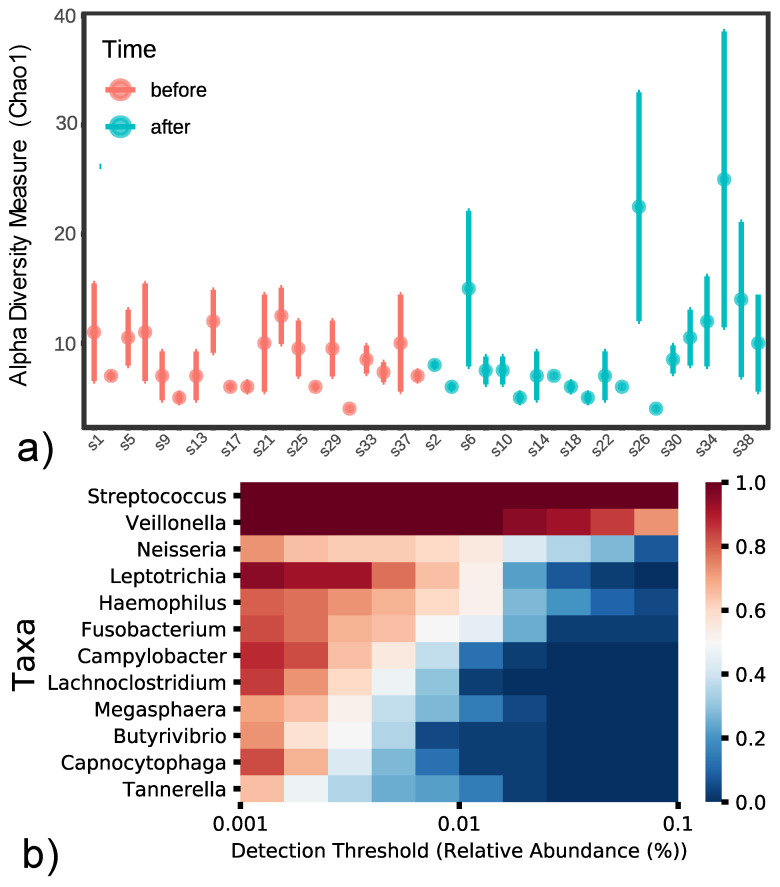
Changes in salivary microbiome before and after anticancer treatment: (**a**) Alpha diversity measures for matched patient samples (PAT(#ID)(before/after)) using Chao1 index (red—before, blue—after); (**b**) Heatmap of taxon prevalence by its relative abundance, cell color represents a relative prevalence of taxa at certain detection threshold (from 0 to 1, blue to red).

**Figure 3 cancers-14-00007-f003:**
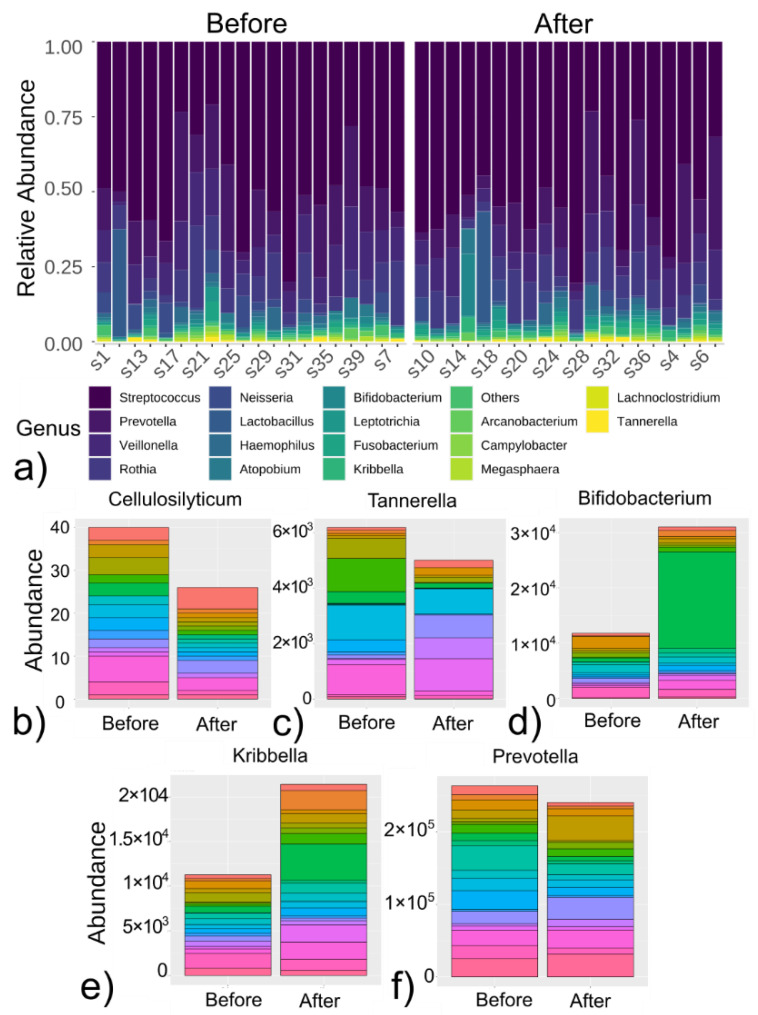
Changes in salivary microbiome before and after anticancer treatment for specific patient: (**a**) paired measurements of relative abundances of genera for a specific patient, before and after anticancer therapy; (**b**–**f**) stacked bar plots for genus relative abundance before and after anticancer therapy. Layer color indicated specific patient samples (across **b**–**f** graphs). The demonstrated genera were significantly different between paired measures (before and after therapy), *p*-value < 0.15 using Wilcoxon signed-rank test.

**Figure 4 cancers-14-00007-f004:**
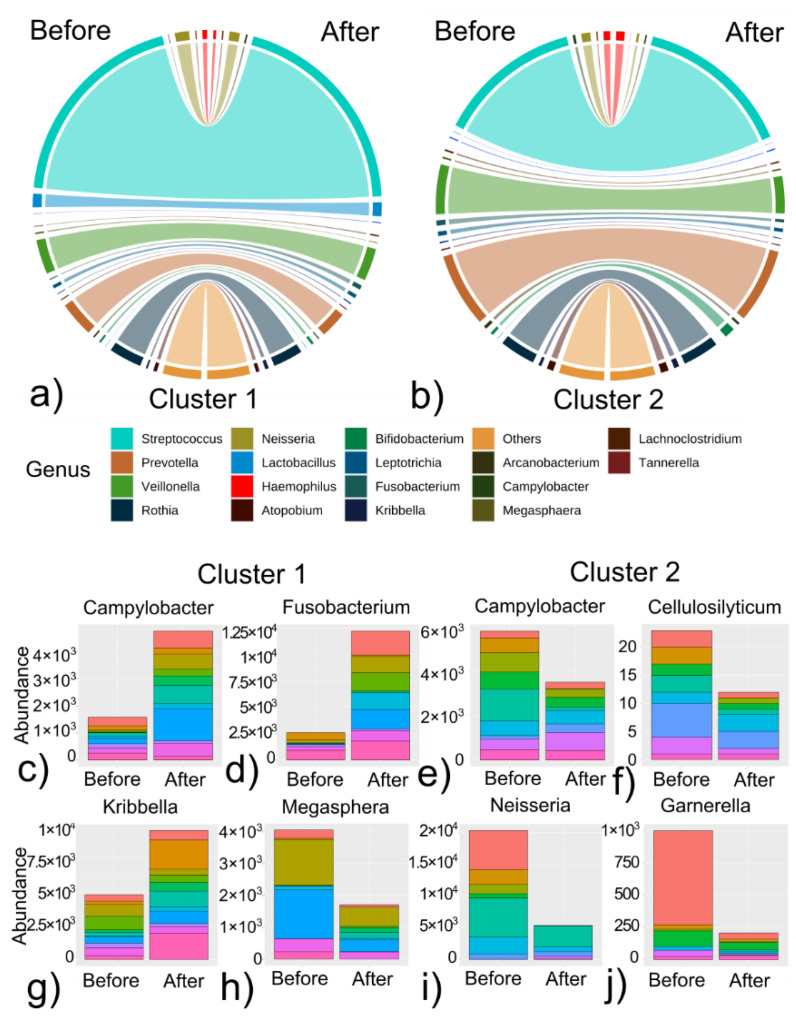
Clustering analysis of microbiome changes after anti-cancer therapy: (**a**,**b**) chord plots represent the relative abundance estimated for a perfect profile in Cluster 1 and Cluster 2. Values before and after therapy are right- and left side of a circle respectively. Colors for both chord plots correspond to the genus (legend “GENUS”); (**c**–**j**). Stacked bar plots represent the difference in relative abundance specific for clusters (Cluster 1 = **c**,**d**,**g**,**h**; Cluster 2 = **e**,**f**,**i**,**j**). Color layers represent patients for a specific group and they are consequently used within a group.

**Figure 5 cancers-14-00007-f005:**
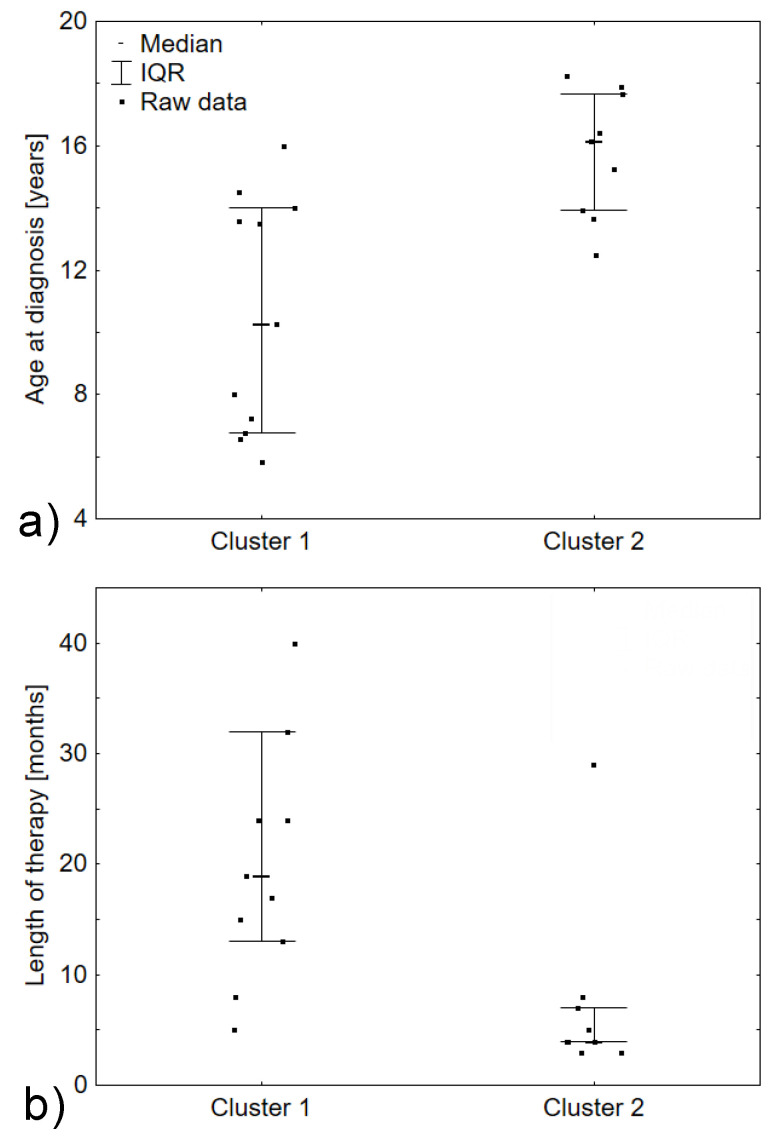
Clinical diversity of identified clusters: (**a**) difference in the age at diagnosis of patients from Cluster 1 and Cluster 2; *p* = 0.0049. (**b**) Difference in the length of anti-cancer therapy between patients from Cluster 1 and Cluster 2; *p* = 0.0038.

**Figure 6 cancers-14-00007-f006:**
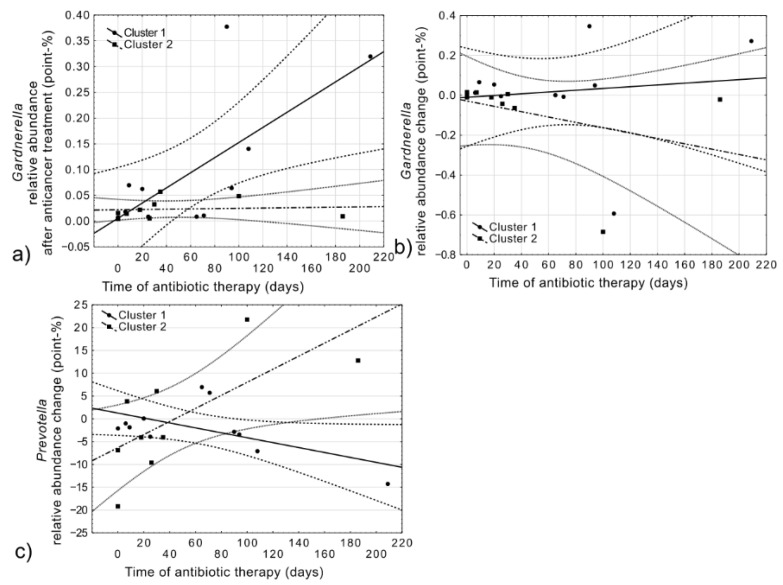
Spearman correlations between duration (time) of antibiotic therapy (days) and relative abundance of *Gardnerella* before (**a**), after (**b**), and *Prevotella* change (**c**) for Cluster 1 (continuous line, circle markers) and Cluster 2 (dashed line, square markers); 95% confidence interval for regression line was marked as dotted lines.

**Table 1 cancers-14-00007-t001:** Clinical characteristics of the study group (*n* = 20).

Clinical Feature	Median (25–75%) or % (*n*)
Age at diagnosis (years)	13.79 (9.13–16.08)
Length of therapy (months)	10.50 (4.50–24.00)
Length of antibiotic therapy (days)	28.00 (8.00–96.50)
Sex (%males)	65% (13/20)

**Table 2 cancers-14-00007-t002:** Comparison of changes in relative abundance before and after the anticancer therapy.

Bacteria	Before Treatment (%)	After Treatment (%)	*p*-Value
*Bifidobacterium*	0.37 (0.28–0.68)	0.72 (0.37–1.00)	0.1005
*Cellulosilyticum*	0.0021 (0.0007–0.0029)	0.0011 (0.0007–0.0012)	0.0438
*Kribbella*	0.48 (0.32–0.68)	0.80 (0.50–1.21)	0.1454
*Prevotella*	12.28 (6.32–17.69)	9.47 (4.93–12.94)	0.1259
*Tannerella*	0.10 (0.05–0.43)	0.09 (0.01–0.27)	0.0366

**Table 3 cancers-14-00007-t003:** Genera abundance summary after unsupervised classification (using k-means), based on patient-specific relative abundances (before and after anticancer therapy). *Campylobacter*, *Fusobacterium*, and *Neisseria* demonstrate a difference in abundance before and after therapy in at least one of the predefined groups. Change in abundance is compared using Wilcoxon signed-rank test, *p* < 0.15 is defined as significant.

Bacteria	Before Treatment (Cluster 1)	After Treatment (Cluster 1)	*p*-Value	Before Treatment (Cluster 2)	After Treatment (Cluster 2)	*p*-Value
*Campylobacter*	0.16 (0.05–0.20)	0.36 (0.20–0.63)	0.0128	0.66 (0.45–0.76)	0.38 (0.29–0.44)	0.1097
*Cellulosilyticum*	0.0014 (0.0000–0.0029)	0.0009 (0.0006–0.0012)	0.5337	0.0029 (0.0021–0.0030)	0.0011 (0.0009–0.0012)	0.0506
*Fusobacterium*	0.09 (0.01–0.33)	1.52 (0.15–1.81)	0.0208	0.85 (0.26–1.61)	0.71 (0.41–0.91)	0.5940
*Gardnerella*	0.01 (0.01–0.03)	0.06 (0.01–0.14)	0.1823	0.03 (0.01–0.05)	0.02 (0.01–0.03)	0.1386
*Kribbella*	0.34 (0.27–0.58)	0.64 (0.45–1.18)	0.1307	0.65 (0.49–0.71)	1.05 (0.69–1.23)	0.1731
*Megasphaera*	0.12 (0.01–0.42)	0.08 (0.01–0.22)	0.0912	0.49 (0.13–0.71)	0.31 (0.16–0.60)	0.5940
*Neisseria*	2.28 (0.08–7.18)	1.21 (0.06–5.40)	0.4769	1.45 (0.69–2.65)	0.09 (0.01–0.73)	0.0284

**Table 4 cancers-14-00007-t004:** Clinical characteristics of the patients according to the cluster adherence.

	Cluster 1	Cluster 2	
Clinical Feature	Median (25–75%) or % (*n*)	Median (25–75%) or % (*n*)	*p*-Value
Age at diagnosis (years)	10.25 (6.75–14.00)	16.16 (13.91–17.66)	0.0049
Length of anticancer therapy (months)	19.00 (13.00–32.00)	4.00 (4.00–7.00)	0.0038
Length of antibiotic therapy (days)	71.00 (9.00–108.00)	26.00 (7.00–35.00)	0.4467
Sex (% males)	63.64% (7/11)	66.67% (6/9)	0.7415
Steroid therapy (Yes)	27.27% (3/11)	66.67% (6/9)	0.1902
Diagnosis: leukemia/lymphoma	45.45% (5/11)	66.67% (6/9)	0.8812
Diagnosis: CNS tumor	27.27% (3/11)	11.11% (1/9)	
Diagnosis: soft tissue tumor	27.27% (3/11)	22.22% (2/9)	
Caries (Yes)	63.64% (7/11)	55.56% (5/9)	0.7136

## Data Availability

The data presented in this study are available upon request from the corresponding author and in Bioproject with Accession: PRJNA770171 https://www.ncbi.nlm.nih.gov/bioproject/?term=PRJNA770171. Last access 12 December 2021.

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
