# Peer review of "Evaluation of Changes to the Oral Microbiome Based on 16S rRNA Sequencing among Children Treated for Cancer"

_cancers, 2021, doi:10.3390/cancers14010007_

Round 1

Reviewer 1 Report

Thank you for accepting my comments and suggestions.

Below I included minor changes to be made on the document.

The term “microflora” continues to appear in the manuscript. Please revise.

The term “S16 rRNA” continues to appear in the manuscript. Please revise.

Although microbiome has been replace in the title and some parts of the text, it still appears in the abstract and throughtout, which is not consistent with the title of the document. Please revise accordingly. If the authors prefer microbiome, it may be used, but please be coherent along the document. You also use microbiome and microbiota as synonyms and that is not entirely true.

Genus Cellulosiluticum is incorrectly spelled in some parts of the document

I’m unfamiliar with the genus Kinetoplastibacterium, do you mean "Candidatus Kinetoplastibacterium"?

Lactobacillus taxonomy has recently been revised. Take this under consideration and confirm ALL taxonomic taxa mentioned on the manuscript for full accuracy.

Reviewer 2 Report

the authors performed all referee 2 comments. The paper is now suitable for the publication

This manuscript is a resubmission of an earlier submission. The following is a list of the peer review reports and author responses from that submission.

Round 1

Reviewer 1 Report

Evaluation of changes to the oral microbiome based on S16 rRNA sequencing among children treated for cancer.

Scope

The subject under study is relevant. The oral microbiome of children, and changes due to cancers treatments, should be addressed to enhance the knowledge on this issue. The incidence of caries is augmented and further information may allow the development of preventive and/or treatment measures.

General comments

The English language (sentence and spelling) must be revised throughout the manuscript.

Specific comments

The title presents and error “S16 rRNA” must be replaced by “16S rRNA”

Simple summary:

“16s” must be replaced by “16S”. The term microflora is outdated.

Abstract:

The abstract is a rather important part of a manuscript, as it has the mission of appealing to the readers. This specific abstract does not comply with those purposes. It lacks a proper introduction to the subject under study. Results are displayed, but not conclusion is withdraw from the data and no prospects of future application of this knowledge are proposed. It must be re-written.

Introduction:

Once again the term outdated “flora” is used. Scientific names lack italic.

There are several recent studies addressing the oral microbiome (https://pubmed.ncbi.nlm.nih.gov/32350240/; https://pubmed.ncbi.nlm.nih.gov/33204698/; https://pubmed.ncbi.nlm.nih.gov/33532491/; https://pubmed.ncbi.nlm.nih.gov/33774891/) that should have been mentioned, not even the human microbiome project, of worldwide importance (https://www.hmpdacc.org/ ). In fact, the authors do not characterize the “normal” oral microbiome of children therefore, the study is not properly introduced.

Material and Methods:

Why did the authors use lysozyme and mutanolysin, but not lysostaphin? S. aureus is frequently present in the oral and perioral regions.

The title of the manuscript uses the term microbiome, but in fact the DNA extraction methodology is directed towards bacteria.

Results and Disction sections:

The term bacteriome is used, although it has not been introduced before. Flora and microflora continue to be used.

Results description is confusing. The authors mention genus allocation but mention Streptococcaceae, Prevotellaceae… The scientific names are not Italianized.

Data is presented using distinct display options, turning interpretation confusing. The most relevant forms should be selected and redundant displays removed.

Were the children submitted to antibiotherapy during cancer treatments (ie, between sample collection)? This is a relevant issue, which must be properly presented and discussed.

The children present distinct types of cancer. Did the treatments differ? How did distinct treatments impact the oral microbiome? Results not presented nor discussed.

Overall, since no “normal” oral microbiome is mentioned, and healthy children were not included, turning it is impossible to establish reliable comparisons. Furthermore, bacteria found to be present in the oral microbiome were not correlated with oral infections, caries prevalence or other related problems. Thus, why was this study performed? Only to gather new data? No follow-up was performed? Did the children develop any oral problems in the subsequent years?

What are the implications of these data? How can this knowledge be applied to benefit the children? Many questions remain to be answered.

Author Response

Dear Editor an Reviewer, 

please find our answers attached in the file. 

Best regards,

Patrycja Proc 

Reviewer 2 Report

In this MS authors follow the changes in the oral bacteriome among the children treated for cancer. Even though the presented data are very interesting and are of great interest to readers, I mean that this study would have been much appreciated if they better conducted their experiment and they reported it in a more acceptable manner. General weaknesses include:

  • Collection of saliva samples should be described in more detail – how long time after breakfast, the mouth was washed with water before collection, the collection was done only one-time. If the collection was done only once, there is a great possibility of collection bias. This point is critical for all other analyses.
  • The NGS data processing and analysis should be described in more detail. Moreover, the data should be deposited in Sequence Read Archive (SRA) or other databases to be freely accessible.
  • In the case of Statistical analysis, some correction for multiple testing was applied to data?
  • It should be interesting to interpret data to specific cancer types - at least 3 patients are responsible. It could bring some other interesting findings.
  • It is not clear if some patients were treated with antibiotics during therapy which has a great impact on their bacteriomes.
  • The authors should present more alpha indices – Shannon or Simpson.
  • The results from beta diversity clustering should be supported by some statistical analysis, like unifrac or others.
  • Interpretation of NGS results could have been treated in a deeper fashion, but at present style what would be eventually updated from previous knowledge?
  • Page 5, line 175 – OTUs are not oral taxes units, it should be corrected.
  • It must be better explained what means colours are indicated in Figure 3b-f and Figure 4c-j.
  • The names of bacteria should be in italics.
  • Too many English errors were there to read down. The authors must do peer-reviewing within themselves with more care prior to submission.

Author Response

please find our answers attached in the file. 

Best regards, 

Patrycja Proc 
